# SHIFTS 2.0: EXTENDING THE DATASET OF REAL DISTRIBUTIONAL SHIFTS

## ABSTRACT

Distributional shift, or the mismatch between training and deployment data, is a significant obstacle to the usage of machine learning in high-stakes industrial applications, such as autonomous driving and medicine. This creates a need to be able to assess how robustly ML models generalize as well as the quality of their uncertainty estimates. Standard ML datasets do not allow these properties to be assessed, as the training, validation and test data are often identically distributed. Recently, a range of dedicated benchmarks have appeared, featuring both distributionally matched and shifted data. The Shifts dataset stands out in terms of the diversity of tasks and data modalities it features. Unlike most benchmarks, which are dominated by 2D image data, Shifts contains tabular weather forecasting, machine translation, and vehicle motion prediction tasks. This enables models to be assessed on a diverse set of industrial-scale tasks and either universal or directly applicable task-specific conclusions to be reached. In this paper, we extend the Shifts Dataset with two datasets sourced from industrial, high-risk applications of high societal importance. Specifically, we consider the tasks of segmentation of white matter Multiple Sclerosis lesions in 3D magnetic resonance brain images and the estimation of power consumption in marine cargo vessels. Both tasks feature ubiquitous distributional shifts and strict safety requirements due to the high cost of errors. These new datasets will allow researchers to explore robust generalization and uncertainty estimation in new situations. This work provides a description of the dataset and baseline results for both tasks.

## 1 INTRODUCTION

In machine learning it is commonly assumed that training, validation, and test data are independent and identically distributed, implying that good testing performance is a strong predictor of model performance in deployment. This assumption seldom holds in real, "in the wild" applications. Real-world data are subject to a wide range of possible *distributional shifts* – mismatches between the training data and the test or deployment data (Quiñonero-Candela, 2009; Koh et al., 2020; Malinin et al., 2021). In general, the greater the degree of the shift in data, the poorer the model performance on it. While most ML practitioners have faced the issue of mismatched training and test data at some point, the issue is especially acute in high-risk industrial applications such as finance, medicine, and autonomous vehicles, where a mistake by the ML system may incur significant financial, reputational and/or human loss.

Ideally, ML models should demonstrate robust generalization under a broad range of distributional shifts. However, it is impossible to be robust to *all* forms of shifts due to the *no free lunch theorem* (Murphy, 2012). ML models should therefore indicate when they fail to generalize via uncertainty estimates, which enables us to take actions to improve the safety of the ML system, deferring to human judgement (Amodei et al., 2016; Malinin, 2019), deploying active learning (Settles, 2009; Kirsch et al., 2019) or propagating the uncertainty through an ML pipeline (Nair et al., 2020). Unfortunately, standard benchmarks, which contain i.i.d training

and held-out data, do not allow robustness to distributional shift and uncertainty quality to be assessed. Thus, there is an acute need for dedicated benchmarks which are designed to assess both properties.

Recently, several dedicated benchmarks for assessing generalisation under distributional shift and uncertainty estimation have appeared. Specifically, the ImageNet and the associated A, R, C, and O versions of ImageNet (Hendrycks et al., 2020; Hendrycks & Dietterich, 2019; Hendrycks et al., 2021), the WILDS collection of datasets(Koh et al., 2020), the Diabetic Retinopathy dataset (Filos et al., 2019) and the ML uncertainty benchmarks (Nado et al., 2021). The Shifts Dataset (Malinin et al., 2021) is also a recent benchmark for jointly assessing the robustness of generalisation and uncertainty quality. It is a large, industrially sourced data with examples of real distributional shifts, from three very different data modalities and four different predictive tasks - specifically a tabular weather forecasting task (classification and regression), a text-based translation task (discrete autoregressive prediction) and a vehicle motion-prediction task (continuous autoregressive prediction). The principle difference of Shifts from the other benchmarks is that it was specifically constructed to examine data modalities and predictive tasks which are not 2D image classification, which is a task-modality combination that dominates the other benchmarks described above.

This work extends the Shifts Dataset with two new datasets sourced from high-risk healthcare and industrial tasks of great societal importance: segmentation of white matter Multiple Sclerosis (MS) lesions in Magnetic Resonance (MR) brain scans and the estimation of power consumption by marine cargo vessels. These datasets constitute distinct examples of data modalities and predictive tasks that are still scarce in the field. The former represents a structured prediction task for 3D imaging data, which is novel to Shifts, and the latter a tabular regression task. Both tasks feature ubiquitous real-world distributional shifts and strict requirements for robustness and reliability due to the high cost of errors. For both datasets we assess ensemble-based baselines in terms of the robustness of generalisation and uncertainty quality.

## 2 BENCHMARK PARADIGM, EVALUATION AND CHOICE OF BASELINES

**Paradigm** The Shifts benchmark views robustness and uncertainty estimation as having *equal* importance. Models should be robust to as broad a range of distributional shifts as possible. However, through the no free lunch theorem, we know that we cant construct models which are guaranteed to be universally robust to all plausible shifts. Thus, where models fail to robustly generalise, they should yield high estimates of uncertainty, enabling risk-mitigating actions to be taken. Robustness and uncertainty estimation become two sides of the same coin and need to be assessed *jointly*.

The Shifts Dataset was originally constructed with the following attributes. First, the data is structured to have a 'canonical partitioning' such that there are in-domain, or 'matched' training, development and evaluation datasets, as well as a shifted development and evaluation dataset. The latter two datasets are also shifted relative to each other. Models are assessed on the joint in-domain and shifted development or evaluation datasets. This is because a model may be robust to certain examples of distributional shifts and yield accurate, low uncertainty predictions, and also perform poorly and yield high estimates of uncertainty on data matched to the training set. Providing a dataset which contains both matched and shifted data enables better evaluation of this scenario. Second, it is assumed that at training or test time *it is not known a priori* about whether or how the data is shifted. This emulates real-world deployments in which the variation of conditions cannot be sufficiently covered with data and is a more challenging scenario than one in which information about nature of shift is available (Koh et al., 2020). In this work we maintain these attributes.

**Evaluation** Robustness and uncertainty quality are jointly assessed via *error-retention curves* (Malinin, 2019; Lakshminarayanan et al., 2017; Malinin et al., 2021). Given an error metric, error-retention curves depict the error over a dataset as a model's predictions are replaced by ground-truth labels in order of decreasing uncertainty. The area under this curve can be decreased either by improving the predictive performance of the model, such that it has lower overall error, or by providing uncertainty estimates which are better cor-

related with error. Thus, the area under the error retention curves (R-AUC) is a metric which jointly assesses robustness to distributional shift and uncertainty quality. More details are provided in appendix B.

**Choice of Baselines** We consider ensemble-based baselines in this work for three reasons. First, ensemble-based approaches are a standard way to *both* improve robustness *and* obtain interpretable uncertainties (Malinin, 2019; Lakshminarayanan et al., 2017; Gal & Ghahramani, 2016; Gal, 2016; Notin et al., 2020). Most robust learning methods do not estimate uncertainty (Arjovsky et al., 2019; Koh et al., 2020; Sagawa et al., 2021) and most uncertainty estimation techniques have little impact on model performance Malinin & Gales (2018); Notin et al. (2020); Liu et al. (2020); van Amersfoort et al. (2021); Mukhoti et al. (2021). Second, ensembles can be applied to any task with little adaptation (Malinin & Gales, 2021; Malinin et al., 2021; Nair et al., 2020; Mehta et al., 2022; Xiao et al., 2019; Filos et al., 2020; Fomicheva et al., 2020). Third, they do not require information about the nature of distributional shift at training time, unlike other robust learning or domain adaptation methods, such as IRM. The main downside of ensembles is their computational and memory cost, but there has been work on overcoming this limitation (Malinin et al., 2020; Ryabinin et al., 2021; Havasi et al., 2020). To our knowledge, there are no other approaches which have all three properties.

## 3 White Matter Multiple Sclerosis Lesion Segmentation

The first dataset focuses on the segmentation of white matter lesions (WML) in 3D Magnetic Resonance (MR) brain images that are due to Multiple Sclerosis (MS). MS is a debilitating, incurable and progressive disorder of the central nervous system that negatively impacts an individual's quality of life. Estimates claim that every five minutes a person is diagnosed with MS, reaching 2.8 million cases in 2020 and that MS is two-to-four times more prevalent in women than in men (Walton et al., 2020). MRI plays a crucial role in the disease diagnosis and follow-up, as it allows physicians to manually track the lesion extension, dissemination, and progress over time (Thompson et al., 2017). However, manual annotations are expensive, time-consuming, and prone to inter- and intra-observer variations. Automatic, ML-based methods may introduce objectivity and labor efficiency in the tracking of MS lesions and have already showed promising results for the cross-sectional and longitudinal analysis of WML (Zeng et al., 2020).

Patient data are rarely shared across medical centers and the availability of training images is limited. No publicly available dataset fully describes the heterogeneity of the pathology in terms of disease severity and progression, reducing the applicability and robustness of automated models in real-world conditions. Furthermore, changes in the MRI scanner vendors, configurations, imaging software or medical personnel using the devices lead variability in image acquisition in terms of voxel resolution, signal-to-noise ratio, contrast parameters, slice thickness, non-linearity corrections, etc. These differences, which are exacerbated when considering images collected from multiple medical centers, represent significant distributional shifts for segmentation models. Models developed in one medical center may transfer poorly to a different medical center, show little robustness to technical and pathological variability and thus yield poor performance. The development of robust MS lesion segmentation models which can indicate when and *where* they are wrong would bring improvements in the quality and throughput of the medical care available to the growing number of MS patients. Ideally, this would allow patients to receive treatment tailored to their unique situations.

**Task Description** White matter MS lesion segmentation involves the generation of a 3D per-voxel segmentation mask of brain lesions in multi-modal MR images (Rovira et al., 2015). Given an input 3D MRI scan, a model classifies each voxel into a lesion or non-lesion tissue. Two standard modalities for MS diagnosis are T1-weighted and, more commonly, Fluid-Attenuated Inversion Recovery (FLAIR) [1].

**Data** The Shifts MS segmentation dataset is a combination of several publicly available and one unpublished datasets (see Table 1). Specifically, ISBI(Carass et al., 2017b;a), MSSEG-1 (Commowick et al., 2018),

---

[1] Such modalities represent information captured by differing configurations of the scanner's magnetic field. In particular, FLAIR highlights the MS lesions as high-contrast regions within the gray-scale image (Wattjes et al., 2021).

PubMRI (Lesjak et al., 2017) and a dataset collected at the university hospital of Lausanne. The latter has not been released for privacy reasons and will be kept as a hidden evaluation set. A permanent leaderboard will be setup on Grand-Challenge and model evaluation will be enabled via dockers. Patient scans come from multiple clinical centers (locations in Table 1): Rennes, Bordeaux and Lyon (France), Ljubljana (Slovenia), Best (Netherlands) and Lausanne (Switzerland). The data from the locations Rennes, Bordeaux and Lyon originate from MSSEG-1; Best from ISBI; Ljubljana from PubMRI.

Each sample in the Shifts MS dataset, detailed in table 1, consists of a 3D brain scan taken using both the FLAIR and T1 contrasts. Scans from different locations vary in terms of scanner models, local annotation (rater) guidelines, scanner strengths (1.5T vs 3T) and resolution of the raw scans. Each sample has undergone standardized pre-processing including denoising (Coupé et al., 2008), skull stripping (Isensee et al., 2019), bias field correction (Tustison et al., 2010) and interpolation to a 1mm isovoxel space. The brain mask is learned from the T1 modality registered to the FLAIR space (Commowick et al., 2012). The ground-truth segmentation mask, also interpolated to the 1mm isovoxel space, is obtained as a consensus of multiple expert annotators and as a single mask for Best and Lausanne.

| Location | Scanner | Mag. Field | Resolution (mm$^3$) | Raters | Trn | Dev$_{in}$ | Evl$_{in}$ | Dev$_{out}$ | Evl$_{out}$ |
|---|---|---|---|---|---|---|---|---|---|
| Rennes | S Verio | 3.0 T | $0.50 \times 0.50 \times 1.10$ | 7 | 8 | 2 | 5 | 0 | 0 |
| Bordeaux | GE Disc | 3.0 T | $0.47 \times 0.47 \times 0.90$ | 7 | 5 | 1 | 2 | 0 | 0 |
| Lyon | S Aera | 1.5 T | $1.03 \times 1.03 \times 1.25$ | 7 | 10 | 2 | 17 | 0 | 0 |
|  | P Ingenia | 3.0 T | $0.74 \times 0.74 \times 0.70$ |  |  |  |  |  |  |
| Best | P Medical | 3.0 T | $0.82 \times 0.82 \times 2.20$ | 2 | 10 | 2 | 9 | 0 | 0 |
| Ljubljana | S Mag | 3.0 T | $0.47 \times 0.47 \times 0.80$ | 3 | 0 | 0 | 0 | 25 | 0 |
| Lausanne | S Mag | 3.0 T | $1.00 \times 1.00 \times 1.20$ | 2 | 0 | 0 | 0 | 0 | 74 |

Table 1: Meta information and canonical splits for the WML dataset. Scanner models are: Siemens Verio, GE Discovery, Siemens Aera, Philips Ingenia, Philips Medical, Siemens Magnetom Trio.

For standardized benchmarking we have created a *canonical partitioning* of the data into in-domain train, development (Dev) and evaluation (Evl) as well as shifted Dev and Evl datasets. Rennes, Bordeaux, Lyon and Best are treated as the in-domain data. Ljubljana and Lausanne are treated as publicly available and heldout shifted development and evaluation sets, respectively. The diverse locations within Europe act as proxies for multiple shifts including scanner model, disease prevalence, ethnicity, acquisition protocol as well as other latent shifts. However, we cannot disentangle these latent shifts further due to a lack of appropriate metadata. The representativeness of the data across ethnicities and populations with strongly varying disease prevalence is an import consideration, which would be valuable to explore when new datasets are being collected. However, this is something which cannot be explored in the current benchmark. When multiple scans are available per patient, we ensure that all scans for a particular patient appear only in one dataset. This partitioning was selected to create a clear shift between the in-domain and shifted data. Refer to Appendix C.1 for details regarding the choice of the splits. Details on the licensing and distribution are provided in appendices A and D. Sub-population statistics are in appendix C.5.

**Assessment** Segmentation of 3D MRI images is typically assessed via the Dice Similarity Coefficient (DSC)(Dice, 1945; Sørensen et al., 1948) between manual annotations and the model's prediction. However, DSC is strongly correlated with *lesion load* (volume occupied by lesion) - patients with higher lesion load will have a higher DSC (Reinke et al., 2021). We consider a normalized DSC (nDSC), described in Appendix C.2.1, that de-correlates model performance and lesion load. Predictive performance is assessed

via average nDSC across all patients in a dataset. Given the nDSC scores, we construct an error-retention curve and calculate the area under the curve to jointly assess uncertainty and robustness [2].

**Methods** Baseline segmentation models are based on the 3D UNET architecture (Çiçek et al., 2016) with hyperparameters tuned according to La Rosa et al. (2020). Specifically, the model is trained for a maximum of 300 epochs with early-stopping based on Dev-in performance. The model relies on splitting the volume into 3D patches of $96 \times 96 \times 96$ voxels; at training time 32 patches are sampled from each input volume with a central lesion voxel while; at inference patches overlapping by 25% are selected across the whole 3D volume with Gaussian weighted averaging for the final prediction of each voxel belonging to multiple patches. The model probabilities for each voxel are thresholded to generate the per-voxel segmentation map. The threshold is tuned on the Dev-in split. A deep ensemble (Lakshminarayanan et al., 2017) is formed by averaging the output probabilities from 5 distinct single UNET models. Monte Carlo dropout (Gal & Ghahramani, 2016) (MCDP) ensembles are also considered a baseline. Here, 5 UNET-DP models are trained with 50% dropout in each model. For MCDP, a single model is taken and dropout is turned on at inference time with an ensemble formed from 5 separate runs of the model. The process is repeated for each of the single models with dropout to get averaged results. Finally, we also consider deep ensembles of UNETR (Hatamizadeh et al., 2021) models, which feature a transformer-based encoder and a convolutional decoder. The training and inference regime for the UNETR is identical to the UNET. As each single model yields a per-voxel probabilities, ensemble-based uncertainty measures(Malinin, 2019; Malinin & Gales, 2021) are available for uncertainty quantification. In this work, all ensemble models use reverse mutual information (Malinin & Gales, 2021) as the choice of uncertainty measure. Single models use the entropy of their output probability distribution at each voxel to capture the uncertainty. All results reported for single models are the mean of the individual model performances.

| Type | Model | nDSC (%) ($\uparrow$) | | | | R-AUC (%) ($\downarrow$) | | | |
|---|---|---|---|---|---|---|---|---|---|
| | | $Dev_{in}$ | $Dev_{out}$ | $Evl_{in}$ | $Evl_{out}$ | $Dev_{in}$ | $Dev_{out}$ | $Evl_{in}$ | $Evl_{out}$ |
| Single | UNET | 68.54 | 49.33 | 67.59 | 55.79 | 2.51 | 7.84 | 2.77 | 9.87 |
| | UNET-DP | 59.73 | 48.35 | 63.93 | 54.43 | 2.62 | 8.76 | 2.66 | 9.71 |
| | UNETR | 71.21 | 51.60 | 69.27 | 56.76 | 1.89 | 6.17 | 1.95 | 6.47 |
| Ensemble | UNET | 69.70 | 50.85 | 68.89 | 57.53 | 1.17 | 4.66 | 1.76 | 7.40 |
| | UNET-DP | 60.65 | 44.70 | 61.78 | 50.06 | 1.92 | 6.77 | 2.52 | 7.89 |
| | UNETR | **72.51** | **53.46** | **71.41** | **59.49** | **0.34** | **1.52** | **0.63** | **2.88** |

Table 2: Segmentation performance (nDSC) and joint eval of robustness and uncertainty (R-AUC).

**Baseline Results.** Table 2 presents voxel-level predictive performance and joint robustness and uncertainty performance of the considered baselines in terms of nDSC and R-AUC, respectively.[3]. Several trends are evident in the results. Firstly, comparing the in-domain predictive performance against the shifted performance, it is clear that the shift in the location leads to severe degradation in performance at the voxel-scale with drops exceeding 10% nDSC. This clearly shows that out benchmark allows discriminating between robust and non-robust models. Secondly, the transformer-based architecture, UNETR, is able to outperform the fully convolutional architecture, UNET, for all models by about 2% nDSC across the various splits. This demonstrates that transformer based approaches are promising for medical imaging, despite the low-data scenario. However, it is also valuable to highlight that even though UNETR yields better performance, the degree the performance degradation is about the same as for the UNET-based models. Third, dropout, as a regularisation technique, adversely affects the UNET architecture and leads to a severe performance

---

[2]Technically, we calculate the above between the curve and a horizontal line at 1, as nDSC is 'accuracy' metric.

[3]Please refer to Table 10 in the appendix for a lesion-level assessment of performance

drop. This seems to suggest that either the current learning procedure is not very stable and additional noise prevents good convergence, or that the 'standard' UNET model for the task is too small, and dropout over-regularizes it. Finally, comparing deep ensembles against single models, it is clear that ensembling, as expected, boosts predictive performance. Notably, the UNETR gain more performance on shifted data from ensembling than UNET models. At the same time, dropout ensembles yield a decrease in performance.

Now let's examine the baselines in terms of joint assessment of robustness and uncertainty. Again, there are a number of observations to be made. Firstly, there is again a clear degradation of performance between the shifted and in-domain data. Secondly, UNETR models, both single and ensembled, yield by far the best performance, which shows that they are both more robust and yield better uncertainties. Thirdly, what is especially notable is that despite inferior predictive performance relative to the single-model counterpart on shifted data (48.35 vs 44.7), the MCDP ensemble yields improved performance in terms of R-AUC (8.76 vs 6.77). This highlights the value in quantifying knowledge (epistemic) uncertainty, and that systems which are less robust can still be competitive by providing informative estimates of uncertainty. Sub-population analysis of results in provided in appendix C.5.

## 4 VESSEL POWER ESTIMATION

The second dataset involves predicting the energy consumption of cargo-carrying vessels in different weather and operating conditions. Such models are used to optimize route of cargo vessels for minimum fuel consumption. Maritime transport delivers around 90% of the world's traded goods (Christodoulou & Woxenius, 2019), emitting almost a billion tonnes of $CO_2$ annually (Hilakari, 2019). Energy consumption varies depending on the chosen routes, speeds, operation and maintenance of ships. The complex underlying relationships are not fully known or taken into account when these decisions are made, leading to fuel waste. Lack of predictability of fuel usage needs leads to vessels carrying more fuel than necessary, costing more fuel to carry. Training accurate consumption models, both for use on their own and for downstream route optimisation, can help significantly reduce costs and emissions (Gkerekos et al., 2019; Zhu et al., 2020).

While performance data is increasingly collected from vessels, data is still scarce and sensors are prone to noise. Weather and sea conditions that affect vessel power are highly variable based on seasonality and geographical location and cannot be fully measured. Further, phenomena such as the accumulation of marine growth on the vessel's hull (hull fouling) cause the relationship between conditions and power to unpredictably shift over time. As a result, significant distributional shifts occur between the real use cases of models and the data used to train and evaluate them. Inaccurate power prediction and the resultant errors in fuel planning and route optimisation can be costly, hazardous and place the vessel, its crew and cargo at risk. In the context of routing and autonomous navigation, inaccurate modeling of speed-power relation can lead to instructed speeds that cause the engine to enter unsafe barred RPM and power zones or the adoption of excessive speeds in extreme weather conditions. In the context of automated bunker planning - if a vessel incorrectly predicts the fuel requirements for a voyage it could run out of fuel in the middle of the ocean. Thus, development of uncertainty-aware and robust power consumption models is essential to enable the safe and effective deployment of this technology to reduce the carbon footprint of global supply chains.

**Task Description** This is a scalar regression task that involves predicting the main engine shaft power, which can be used to predict fuel consumption given an engine model, at a particular timestep based on tabular features describing vessel and weather state. A probabilistic model would yield a probability density over power consumption. The target power is mostly attributed to the current timestep, but due to transient effects (e.g inertia) and hull fouling prior timesteps can also affect target power.

**Assessment** Predictive performance is assessed using Root Mean Square Error (RMSE), Mean Absolute Error (MAE) and Mean Absolute Percentage Error (MAPE). Area under Mean Square error (MSE) and F1

retention curves (Malinin, 2019; Malinin et al., 2021) is used to assess jointly the robustness to distributional shift and uncertainty quality. The respective performance metrics are named R-AUC and F1-AUC.

**Data** The Shifts vessel power estimation dataset consists of measurements sampled every minute from sensors on-board a cargo ship over 4 years, cleaned and augmented with weather data from a third-party provider. Noise in the data arises due to sensor noise, measurement and transmission errors, and noise in historical weather. Distributional shift arises from hull fouling, sensor calibration drift, and variations in routes and non-measured sea conditions such as water temperature and salinity, which vary across regions and seasons. The features are detailed in Appendix D.2. Licensing and distribution are described in appendices A and D.

In addition we provide a synthetic dataset created using an analytical physics-based vessel model. The synthetic data contains the same input features as real data, but the target power labels are replaced with the predictions of a physics model. As vessel physics are well understood, it is possible to create a model which captures most relevant factors of variation. However, this physics model is still a simplified version of reality and therefore is an easier task with fewer factors of variation than the real dataset. It assumes that the dataset features are a *sufficient* description of all relevant factors of variation, which may not be the case. A significant advantage of the physics model is that it allows generating a *generalization dataset* which covers the *convex hull* of possible feature combinations (Figure 1a,b). Here, data is generated by applying the model to input features independently and uniformly sampled from a predefined range. Thus, models can be assessed on both rare and common combinations of conditions, which are equally represented. For example, vessels are unlikely to adopt high speeds in severe weather, which could lead ML models to learn a spurious correlation. This bias would not be detected during evaluation on real data as the same correlation would be present. Evaluating a candidate model on the generalization set assesses its ability to properly disentangle causal factors and generalise to unseen conditions. This generalisation set enables assessing model robustness with greater coverage, even if on a simplified version of reality.

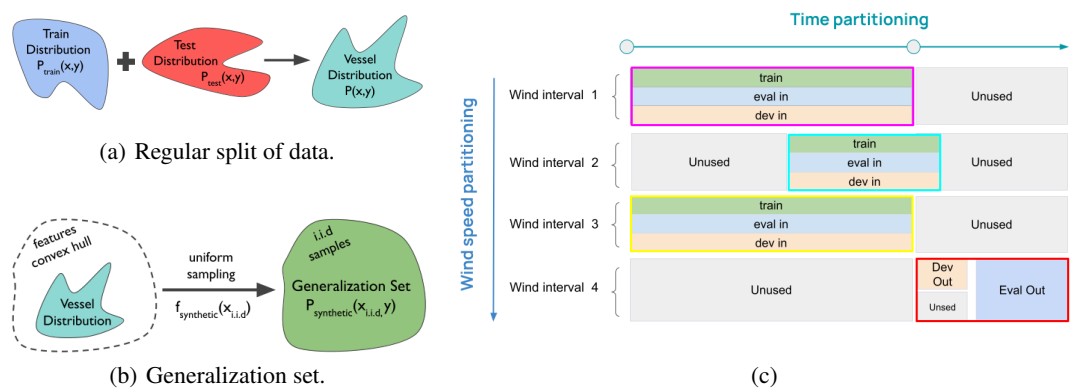

(a) Regular split of data.

(b) Generalization set.

(c)

Figure 1: Partitioning for vessel power dataset. Wind intervals represent 0-3, 3-4, 4-5 and >5 on the Beaufort scale. Train, dev and eval sets contains 530706, 18368 and 47227 records.

For standardized benchmarking, we have created a *canonical partition* of both the real and synthetic datasets (which only differ in targets) into in-domain train, development and evaluation as well as shifted development and evaluation splits. The data is partitioned along two dimensions: wind speed and time, as illustrated in Figure 1c. Wind speed is a proxy for unmeasured components of the sea state, while partitioning in time captures effects such as hull fouling and sensor drift. Additionally, we provide a the synthetic generalisation set of 2.5 millions samples. Real-world performance and robustness to unseen latent factors is assessed using the real dataset, while the synthetic generalisation assesses generalisation and causal disentanglement. The

canonical partitions allow establishing common ground between the real and synthetic data. The best models have high performance on both sets. Models which perform well on the generalisation set and poorly on real data are not robust to unseen latent factors. Conversely, models that perform poorly on the generalisation set and well on real data are strongly affected by spurious correlations in the measured features.

**Methods** We examine the following range of baseline models: Deep Neural Networks (DNNs), Monte-Carlo Dropout ensembles of DNNs, variational DNNs and a proprietary symbolic model. We consider deep ensembles of 10 of each model. In all cases, each ensemble member predicts the parameters of the conditional normal distribution over the target (power) given the input features. As a measure of uncertainty use the total variance - the sum of the variance of the predicted mean and the mean of predicted variance across the ensemble (Malinin, 2019). Baseline methods are detailed in Appendix D.3.

**Baseline Results** Table 3 presents the results of evaluating baseline models on both the real and synthetic versions of the power estimation data. Several trends can be observed. First, the results show that on the proposed data split the shifted data is more challenging for the models to handle - both the error rates R-AUC are higher on the shifted partitions. Secondly, it is clear that the real dataset is overall more challenging than the synthetic dataset for all models, which is expected, as reality contains more unknown factors of variation. Third, of the single model approaches, the variational inference (VI) model consistently yields both the best predictive performance and the best retention performance on all of the real and synthetic canonical partitions. However, on the synthetic generalisation set, which uniformly covers the convex hull of possible inputs, the proprietary symbolic model does best in terms of predictive quality and second-best in terms of R-AUC. Note, the symbolic model yields worst performance RMSE and R-AUC on real data. This highlights the value of the generalisation set - to show which models are overall more robust, rather than just performing well on more typical events in standard train/dev/eval splits. Fourth, the ensembles consistently outperform the mean performance of their single seed counterparts. The ensemble-based results show similar a similar story to those of single models. Here, the ensemble VI model is comparable to or better than other ensemble models on all splits except for the generalisation set, where the symbolic ensemble model does best. Overall, the results show that the ensemble VI models achieves the best overall balance of robustness and uncertainty.

| Method | Model | RMSE (kW) ↓ | | | | | R-AUC ($10^5 kW^2$) ↓ | | |
| --- | --- | --- | --- | --- | --- | --- | --- | --- | --- |
| | | Synthetic | | | Real | | Synthetic | | Real |
| | | In | Out | Gen | In | Out | Full | Gen | Full |
| Single | DNN | 1084 | 1116 | 1487 | 1296 | 1985 | 4.49 | 5.27 | 10.97 |
| | MC dropout | 1078 | 1122 | 1526 | 1271 | 1954 | 4.54 | 5.39 | 10.00 |
| | VI | **1072** | **1109** | 1458 | **1255** | **1916** | **4.33** | **4.53** | **9.57** |
| | Symbolic | 1120 | 1137 | **1213** | 1403 | 2366 | 5.13 | 4.55 | 17.51 |
| Ens. | DNN | 1076 | **1099** | 1427 | 1264 | 1928 | 4.32 | **4.20** | 9.52 |
| | MC dropout | **1069** | 1111 | 1498 | 1248 | 1925 | 4.47 | 4.97 | 9.28 |
| | VI | **1069** | 1104 | 1446 | **1243** | **1895** | **4.29** | 4.32 | **9.13** |
| | Symbolic | 1117 | 1133 | **1204** | 1393 | 2341 | 5.09 | 4.41 | 13.56 |

Table 3: Results on the real and synthetic canonical eval partitions and on the generalization set.

Figure 4 reveals additional insights. Figure 4a shows that joint uncertainty and robustness performance of all models on the full (in+out) real and synthetic evaluation sets are strongly correlated. Figure 4b shows that there is a trade-off between broad robustness and high performance on the real data. The symbolic model, a low variance, high bias models, isn't great overall, but it doesn't fail as strongly as the neural models in unfamiliar situations. Conversely, neural models (low bias, high variance models) are better able to exploit the correlations within the real data, but feature brittle generalisation. Finally, Figure 4c shows the benefits of uncertainty quantification. While the neural models are consistently worse than the symbolic model on the generalisation set, they show comparable or superior joint robustness and uncertainty (R-AUC) to the

symbolic models. Thus, using uncertainty estimates to detect errors, the neural models can achieve superior operating points to the symbolic models. Additional results are provided in Appendix D.4.

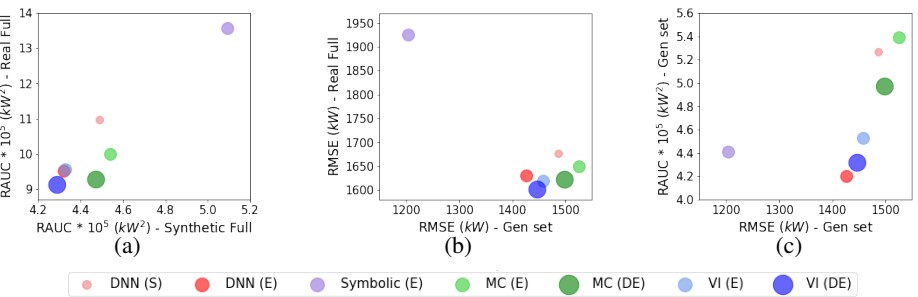

Figure 2: Power estimation key performance trends.

## 5 DISCUSSION AND CONCLUSION

In this paper we extended the Shifts Dataset (Malinin et al., 2021) with two datasets sourced from industrial, high-risk applications of high societal importance which feature ubiquitous distributional shift. These new datasets present novel conditions in which to explore robustness and uncertainty estimation, which we hope will enable additional insights to be drawn. The WML segmentation dataset brings a new modality, a new predictive task and a very low-data regime, where models are almost always operating under some degree of distributional shift even on nominally matched data. This is different from the datasets present in the original release of Shifts, which operated in a large-data regime. The marine cargo-vessel dataset brings another tabular regression task to the table. Being similar to the Shifts weather forecasting dataset Malinin et al. (2021), which is also a tabular regression task, there are key differences. First, while the features are 'tabular', the data described is entirely different - effectively a modality unto itself. Secondly, the fact that naval physics are sufficiently well understood that a reliable physics simulation can be created and used to assess model robustness and generalisation. The real data tests robustness to a distributional shift in the presence of real noise and unseen factors of variation, while the synthetic data enables evaluation of the models' ability to broadly generalize and disentangle causal factors. This is a novel feature of Shifts 2.0 which does not appear in any other benchmarks.

It must be stated that the current benchmark still has limitations. In this work we assess how well uncertainty estimates correlate with the degree of error. Theoretically, uncertainty estimates can enable errors to be detected and risk mitigating actions taken. However, our benchmark does not assess the use of uncertainty for risk mitigating actions or communication of critical information in downstream applications. The main difficulty is that there is a limited consensus on *how* uncertainty can be used in any particular application. For example, we believe that the communication of the uncertainty level of an AI system in healthcare applications is crucial. Uncertainty can, on one side, support the rapid trust calibration of the system and, on the other side, speed up the MRI assessment by attracting the attention of experts to the most uncertain areas [4]. However, it is unclear what is the best way to convey voxel-level uncertainty to a clinician such that it is an asset, rather than a distraction. Furthermore, it is complicated to design a clinical trial that will assess the effectiveness of the utilisation of uncertainty maps for semi-automatic segmentation. While we do not have answer to these questions, we believe that there is value in making them explicit - answering them will be the next step once models which are robust to distributional shift and which yield uncertainty estimates correlated with the likelihood of errors become widely available.

---

[4] See appendix C.4 for an example

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
