# OpenReview forum: "Shifts 2.0: Extending The Dataset of Real Distributional Shifts"
_ICLR.cc/2023/Conference — Submitted to ICLR 2023_

### Official Review · Reviewer_AFoe · 2022-10-25

**Confidence:** 3
**Correctness:** 4
**Technical Novelty And Significance:** 2
**Empirical Novelty And Significance:** 3
**Recommendation:** 6

**Clarity, Quality, Novelty And Reproducibility:**

- The paper is very well-written and clear.
- The application data provided  is not necessarily novel and not the first.
- There is no reproducibility concern as they (will) provide open access data and leader-board.


**Strength And Weaknesses:**

**Strength:**
- Any well maintained dataset is very welcome and needed by the community.
- The dataset/task that has been suggested are also important and relevant,  as the importance of dataset shift in higher-risk application is becoming more clear for the AI community.
- The paper is clearly written, and provided a detailed appendix.

**Weaknesses:**
- The author suggests a permanent leaderboard on grand-challenge as the evaluation set for the brain MRI task, and mentioned it will be possible to evaluate models on it via dockers. No details for further exploration or readiness of such a system are provided to be reviewed by reviewers.
- Using a docker for evaluation can become challenging very fast. I personally prefer .csv submission and I think their are more convenient as it has been used by WILDS leaderboard (https://wilds.stanford.edu/submit/#making-a-submission).
- All of the MRI data is collected in Europe so, there is neither ethnicity related nor disease prevalence shift in the proposed data, and major acquisition or protocol shift would not be observed. Only the scanner type has been changed from one setting to another. However, at least for developing an effective medical image analysis system disease prevalence shift and shifts related to ethnicity are far more important and less explored.
- There is no discussion around patient attribution and metadata related to fairness. Specifically for the medical applications, and in a new setting, one major concern is methods fairness and running the proper sub-group analysis is required. Without any relevant meta-data such as ethnicity this is not feasible. [There is a limited discussion around gender/age and tumor size change between sets which is not sufficient and the paper does not provide any effective insight or evaluation.]


**Summary Of The Paper:**

The paper expands the recently proposed Shifts Dataset with two datasets sourced from high-risk applications. They consider the tasks of segmentation of white matter Multiple Sclerosis lesions in 3D magnetic resonance brain images and the estimation of power consumption in marine cargo vessels.

**Summary Of The Review:**

Contributions around creating of well-maintained and large-scale distribution shift dataset are important and largely welcomed within the community, however, at least for the medical application side, it lacks further evaluation and expansion to provide subgroup meta-data details and addresses concerns related to fairness. Indeed most of the concerns in such a novel and shifted setup are around fairness and performance of this method in sub-groups and long-tails.

---

> ### Author Response · Authors · 2022-11-06
> **Response**
>
> Hi Reviewer 3,
>
> Thank you for your positive review. Regarding your concerns:
>
> - As we said to reviewer cD8F, this system is ready and running, but we didn't mention it further or provide a link due to the anonymity constraint. In fact, the data, code and baseline models are all provided and publicly available.
> - We totally agree that a CSV would be easier. Unfortunately this is not an option for the medical data, especially the data from Lausanne, to which we can provide access, but which we cannot share. To make it easier for users, we have provided detailed step-by-step tutorials on Medium and on GrandChallenge about how to replicate our baselines and create a dockerized submission on grand challenge.
> - Unfortunately, we do not have access to information about the ethnic background of patients - we are not the original data owners / dataset collectors and simply do not have access to this metadata (if it was ever collected) . The metadata that we do have access to are medical centre location and scanner models at those locations. We believe that the diverse locations within Europe (Lausanne, Best, Ljubljana, Lyon, Bordeaux and Rennes, detailed in table 1) act as proxies for multiple shifts including scanner model, disease prevalence (which varies across European countries), ethnicity (which also varies considerably across these locations), acquisition protocol as well as other latent shifts. As we cannot disentangle these shifts due to a lack of appropriate metadata, we made the cleanest cut we could by using location as proxy for numerous shifts. We agree that the representativeness of the data across ethnicities and population with strongly varying disease prevalence is an import consideration,  and we will add a greater discussion of this issue to the paper.
> - We are currently examining whether age and gender metadata is available and can be obtained for each patient in the combined dataset (rather than a group statistic). In the event that we do obtain this metadata, which analyses would you suggest we run? Performance breakdown by age and gender? Or the age / gender balance statistics of the splits we have suggested? Perhaps something else? What would you consider to be thorough sub-group analysis in this context?

---

> > ### Comment · Reviewer_AFoe · 2022-11-15
> > **subgroup analysis**
> >
> > Thanks for the response and clarifications.
> >
> > In case of access to more metadata, reporting exact statistics and any imbalance in each subgroups, and consequently performance breakdown per subgroups are reasonable next steps.

---

> > > ### Author Response · Authors · 2022-11-18
> > > **Subpopulation Analysis**
> > >
> > > Hello!
> > >
> > > So we've managed to get the sub-population analysis done and updated the paper. The sub-population analysis is provided in appendix C.5 of the supplementary (file also updated). We have also added very short discussion of how we have selected the splits and that location is a proxy for multiple shifts, not just scanner type. This is added to the paragraph below table 1 in the main paper.
> > >
> > > ------------
> > > ### Sub-population analysis.
> > >
> > > We examine the gender, age and lesion load distributions across the different splits, and how each of these properties interacts with model performance. As figure 9 shows, in all the datasets, there are roughly four times as many female patients as male patients. This overall is indicative of the higher incidence of MS in women than in men, and that women are more likely to get diagnosed. Female patients, on average, are 2-6 years older than male patients in all datasets, except eval_out, where the female patients are on average younger. Patients in the shifted datasets are on average younger. Patients from Lausanne (eval_out) have a far lower lesion load, as they are in an earlier stage of MS. There are no clear gender and age related correlations with lesion load.
> > >
> > > We have provided a performance break-down by gender for each datasets, as well as performance breakdown by age and lesion load, for both UNet and UNEtr models in figures 10 and 11, respectively. The results show that there are no clear gender-based differences in model performance. Similarly, there is no clear correlation between model performance and age. There is a minor correlation (0.46-0.47) between model performance and lesion load on datasets featuring low lesion load. This is, however, expected, as it shows the intrinsic difficulty of the task. Very small, hard-to-detect lesions are harder to segment accurately. Tables 14 and 15 show the mean, median, minimum, and maximum per-patient nDSC for male and female patients across all datasets. The results on in-domain data show that the performance on male and female patients is similar. Out of domain, performance on female patients is a little lower on average. However, it is important to highlight that these results and statistics are collected based on a small sample size and are therefore noisy. Furthermore, there are 4 times as few male patients than female patients, so statistics on male patients are noisier. Thus, we would be hesitant to make any strong statements regarding sub-population performance bias of our models.
> > >
> > > -------------------------
> > >
> > > We hope that you find this analysis satisfactory. While it is unlikely we will be able to add more during this rebuttal, we would be happy to carry out any additional analyses which you suggest.
> > >
> > > Kind Regards,
> > > Authors.

---

> > ### Comment · Reviewer_AFoe · 2022-11-19
> > **Thanks and Acknowledge**
> >
> > I just wanted to thank the authors for their effort and acknowledge that I have reviewed their answers.

---

### Official Review · Reviewer_cD8F · 2022-10-26

**Confidence:** 3
**Correctness:** 4
**Technical Novelty And Significance:** 3
**Empirical Novelty And Significance:** 3
**Recommendation:** 6

**Clarity, Quality, Novelty And Reproducibility:**

The paper is well-written with all the necessary dataset details mentioned in the appendices. In terms of reproducibility, the authors have not released code for the baseline models. They are encouraged to release the code. But, the appendices have enough details to reproduce the results.

**Details Of Ethics Concerns:**

The paper introduces two datasets that contain data of high-societal impact. Specifically, they have human subjects' brain MRI scans for MS lesion detection and power consumption by marine vehicles.  Both datasets feature ubiquitous real-world distributional shifts and strict requirements for robustness and reliability due to the high cost of errors (significant financial, reputational, and/or human loss). The authors have addressed this under a section titled: "Guidelines for Ethical Use" in the Appendix.

**Strength And Weaknesses:**

Strength(s):
1. The reasons for creating data splits are sound and well-grounded. They also serve the purpose of demonstrating a realistic distributional shift.
2. The motivation for creating and releasing the datasets is clear and sound. The paper also discusses the potential applications/use cases of the datasets.
3. The datasets are well documented with all the necessary details including license and user agreements.

Weakness(es)/Suggestion(s):
1. It would be nice to compare the brain MRI dataset with existing MS lesion dataset(s) (like MICCAI's MS dataset). Similarly, if available, please do the same for the power consumption tabular dataset.
2. Although it is good to have an exhaustive appendix, it would be a little hard for the readers to switch back and forth between the main text and the appendix. I'm not sure if the appendix can be part of the main paper. If it can, it would ease reading as the hyperlink to appendices would work.
3. Since the authors are hosting a Grande Challenge with a leaderboard, would the evaluation criteria be the same as the one's mentioned in Table 2 (for the segmentation dataset) and Table 3 (for the tabular dataset)?

**Summary Of The Paper:**

The paper introduces two new datasets having distribution shifts to an existing collection of datasets. The two new datasets are white matter Multiple Sclerosis lesions in 3D magnetic resonance brain images and the estimation of power consumption in marine cargo vessels. They also have baseline results for each dataset which helps to evaluate the model's performance in terms of robustness and uncertainty estimation.

**Summary Of The Review:**

In summary, since the paper introduces two well-documented datasets which help to evaluate ML algorithms' robustness and uncertainty estimation capabilities, I think it would be a valuable asset to the community. Therefore, my recommendation would be marginally above the acceptance threshold (6).

---

> ### Author Response · Authors · 2022-11-06
> **Response to Reviewer cD8F**
>
> Hi Reviewer cD8F,
>
> Thank you for your positive review. Regarding your concerns:
>
> - Ok, so there's two bits to this.
>   - Firstly, we’re sorry if this wasn’t clear enough - The Shifts MS segmentation dataset was formed by combining all publicly available MS lesion segmentation datasets that we are aware of, including one of MICCAI’s MS dataset (MSSEG1) . The second MICCAI MS dataset (MSSEG2) cannot be directly compared to or added to the Shifts benchmark, as the task is subtly different - it involves segmenting _new_ lesions, rather than _all_ lesions. Thus, we unfortunately can't make a comparison of model performance across datasets, as we are already making use of all publicly available MS datasets (that we know of). Regarding the vessel power dataset, it is to our knowledge unique data that is the first of its kind.
>   - Could you please be a little more specific on what are you actually looking for in such a comparison? Model performance? Dataset statistics? Perhaps something else?
>
> - We fully appreciate your frustration. The paper was written as a single file. The appendices were split out in order to comply with the ICLR requirements. We were being extra careful, as we heard of past cases when people were desk rejected for having the appendices in the main and similar issues. We will see whether it is possible to upload everything as a single file on OpenReview for convenience.
>
> - Yes, the criteria on grand challenge are the same as in tables 2 and 3.
>
> - Re: Reproducibility - Links to the repository containing the code, the Zenodo repositories for the data, as well as the GrandChallenge leaderboards are already public. We are unable to share them with you within this review due to the anonymity constraint, but they are readily discoverable to anyone who searches for them.
>
> Other than answering the above questions and resolving the question about model comparison, is there anything else that you would wish to see improved in this paper which we can provide in the constraints of this rebuttal that could convince you to raise your appraisal of our work?

---

> > ### Comment · Reviewer_cD8F · 2022-11-18
> > **Thank you for your responses**
> >
> > Thank you authors for the responses.
> >
> > "Could you please be a little more specific": I think you answered my question in the first sub-bullet of the first main bullet.
> >
> > The authors have addressed my concerns and I will keep my score of 6 (marginally above the acceptance threshold) unchanged.

---

### Official Review · Reviewer_1V51 · 2022-10-29

**Confidence:** 3
**Correctness:** 3
**Technical Novelty And Significance:** 2
**Empirical Novelty And Significance:** 2
**Recommendation:** 5

**Clarity, Quality, Novelty And Reproducibility:**

The clarity of this paper is clear and the authors clearly present the motivation of this work, the datasets, and the evaluation procedures. While the novelty (empirical novelty) is limited compared with previous shifts datasets work.

**Strength And Weaknesses:**

Strength:
1. Building a benchmark to study distribution shift is an important topic and introducing the medical image and real industrial problem into this benchmark is useful.
2. The authors propose to evaluate both the robustness and uncertainty is reasonable for me.

Weaknesses:
1. After reading this paper, I am not very clear about the contribution of this paper (including both empirical novelty and technical novelty), especially compared with the previous work shifts Datasets. For me, the two new things are introducing two new datasets and introducing the concept of "both evaluating accuracy and robustness." However, the medical image dataset is a public dataset, and there are also several similar works to study "domain generalization" for medical images, like cardiac image segmentation (https://www.ub.edu/mnms/ and https://www.ub.edu/mnms-2/). For the second one, as the authors mentioned, this dataset is very similar to the weather prediction one in the original shifts Datasets.  It seems that there are not too many "new things" in this paper.

2. The authors want to benchmark the current ML algorithms assessing robustness and uncertainty, while it seems that the author only evaluates some baseline methods, not comprehensively comparing them with various methods. The major contribution is limited to introducing these two different datasets.

3. For the evaluation of uncertainty, I am not clear whether it is really "uncertainty." If my understanding is correct, it is to evaluate the prediction probability (or something like that). I am not sure if it is proper to mention that as "uncertainty."

4. Some previous uncertainty estimations works may need to be mentioned as related work.


**Summary Of The Paper:**

This paper extends the previous Shifts Dataset with two datasets sourced from industrial, high-risk applications of high societal importance which feature ubiquitous distributional shift.  The authors also benchmark some baseline algorithms about model accuracy and robustness with these two different datasets.

**Summary Of The Review:**

In summary, although the studied problem is important, the current version may not have enough contribution (including both empirical and technical novelty) to be accepted at ICLR.

---

> ### Author Response · Authors · 2022-11-07
> **Response to Reviewer 1V51 part 1**
>
> Hello!
>
> Thank you for your thought provoking review! To address your concerns:
>
> 1. You raise an important point. We felt that the response is important to all reviewers as well as yourself. Please see general response 1 above. Additionally, we have a question for you regarding your assessment of the vessel power estimation dataset. The introduction of the synthetic data in addition to the real data for the vessel prediction task enables examining causal disentanglement as distinct from generalisation, by sampling the full convex hull of the data. If you look at the results on the behaviour of different models on the synthetic vs real tasks and the three-way trade-offs between generalisation, uncertainty estimation and causal disentanglement, this task provides a promising new dimension on which to evaluate models. Is there a reason you didn't consider this as a contribution?
>
> 2. As above, this is a datasets paper and benchmarking all current ML algorithms was not its primary focus. As explained in section two of the paper, we chose ensemble methods because they both improve robustness and yield interpretable uncertainty estimates and thus naturally fit within our assessment paradigm. No other methods, to our knowledge, do both.

---

> > ### Author Response · Authors · 2022-11-07
> > **Response to Reviewer 1V51 part 2**
> >
> > 3.  I’m afraid that your interpretation is not entirely accurate here. Please allow us to explain: typically, uncertainty is assessed by obtaining a scalar value which represents how uncertain a model is in it’s prediction. This numerical value should be high in situations where the model is likely to make a mistake - such as when the input is out-of-domain (distributionally shifted) and the model is likely to make a nonsensical prediction, or when it the input is inherently ambiguous or noisy. This scalar value can be the probability of the predicted class, or the entropy of the model’s output distribution, or some measure of ensemble diversity (such as mutual information), or some measure of variance in the case of regression tasks. For structured prediction tasks, such as segmentation or autoregressive prediction, the situation is a little more complex, but the overall idea is the same. This is the established approach to uncertainty in machine learning literature:
> >
> > - Yarin Gal, Uncertainty in Deep Learning, Ph.D. thesis, University of Cambridge, 2016.
> > AndreyMalinin,UncertaintyEstimationinDeepLearningwithapplicationtoSpokenLanguage Assessment, Ph.D. thesis, University of Cambridge, 2019.
> > - Yarin Gal and Zoubin Ghahramani, “Dropout as a Bayesian Approximation: Representing Model Uncertainty in Deep Learning,” in Proc. 33rd International Conference on Machine Learning (ICML-16), 2016.
> > B. Lakshminarayanan, A. Pritzel, and C. Blundell, “Simple and Scalable Predictive Uncertainty Estimation using Deep Ensembles,” in Proc. Conference on Neural Information Processing Systems (NIPS), 2017.
> > - Arsenii Ashukha, Alexander Lyzhov, Dmitry Molchanov, and Dmitry Vetrov, “Pitfalls of in- domain uncertainty estimation and ensembling in deep learning,” in International Conference on Learning Representations, 2020.
> > - L. Smith and Y. Gal, “Understanding Measures of Uncertainty for Adversarial Example Detection,” in UAI, 2018.
> > - Andrey Malinin and Mark Gales, “Uncertainty estimation in autoregressive structured prediction,” in International Conference on Learning Representations, 2021.
> > - Yen-Chang Hsu, Yilin Shen, Hongxia Jin, and Zsolt Kira, “Generalized odin: Detecting out-of-distribution image without learning from out-of-distribution data,” 2020.
> > - Joost Van Amersfoort, Lewis Smith, Yee Whye Teh, and Yarin Gal, “Uncertainty estimation using a single deep deterministic neural network,” in International Conference on Machine Learning. PMLR, 2020, pp. 9690–9700.
> > - Marton Havasi, Rodolphe Jenatton, Stanislav Fort, Jeremiah Zhe Liu, Jasper Snoek, Balaji Lakshminarayanan, Andrew M. Dai, and Dustin Tran, “Training independent subnetworks for robust prediction,” 2020.
> > - Jeremiah Zhe Liu, Zi Lin, Shreyas Padhy, Dustin Tran, Tania Bedrax-Weiss, and Balaji Laksh- minarayanan, “Simple and principled uncertainty estimation with deterministic deep learning via distance awareness,” arXiv preprint arXiv:2006.10108, 2020.
> > - Joost van Amersfoort, Lewis Smith, Andrew Jesson, Oscar Key, and Yarin Gal, “Improving deterministic uncertainty estimation in deep learning for classification and regression,” arXiv preprint arXiv:2102.11409, 2021.
> > - Andrey Malinin and Mark Gales, “Predictive uncertainty estimation via prior networks,” in Advances in Neural Information Processing Systems, 2018, pp. 7047–7058.
> > - Andrey Malinin, Sergey Chervontsev, Ivan Provilkov, and Mark Gales, “Regression prior networks,” 2020.
> > - Andrey Malinin, Bruno Mlodozeniec, and Mark JF Gales, “Ensemble distribution distillation,” in International Conference on Learning Representations, 2020
> > - Angelos Filos, Sebastian Farquhar, Aidan N. Gomez, Tim G. J. Rudner, Zachary Kenton, Lewis Smith, Milad Alizadeh, Arnoud de Kroon, and Yarin Gal, “A systematic comparison of bayesian deep learning robustness in diabetic retinopathy tasks,” 2019.
> > - Andrey Malinin, Liudmila Prokhorenkova, and Aleksei Ustimenko, “Uncertainty in gradient boosting via ensembles,” in International Conference on Learning Representations, 2021.
> > - Angelos Filos, Panagiotis Tigkas, Rowan McAllister, Nicholas Rhinehart, Sergey Levine, and Yarin Gal, “Can autonomous vehicles identify, recover from, and adapt to distribution shifts?,” in International Conference on Machine Learning. PMLR, 2020, pp. 3145–3153.
> >
> > In our work, for the MS lesion segmentation task, we have used the average entropy of the per-voxel output distributions for single segmentation models, and reverse-mutual information for ensembles of segmentation models. For the vessel power prediction task, we have used the predicted variance for individual models and total variance (mean predicted variance + variance of the means) for ensemble of models. This usage is consistent with the established uncertainty estimation literature.
> >
> > 4. We’re happy to add a small discussion of uncertainty estimation. We’ll cite the from the above list of works. We'd be more than happy to add anything else you think is relevant.

---

> > > ### Comment · Reviewer_1V51 · 2022-11-18
> > > **Thanks for your quick update**
> > >
> > > 1. For the first point, I underestimated the importance of the second dataset. I agree that the introduction of synthetic data in addition to the real data can be regarded as a contribution. However, as mentioned in the general response, I do not think collecting publically available medical image segmentation datasets together can be regarded as a major contribution and can meet the criterion of ICLR.
> > >
> > > 2. As a clarification to other reviewers and AC, I am not requesting benchmarking **ALL** ML algorithms. As a benchmark paper, it is important to benchmark the important algorithms in this field. However, the current paper only benchmarks UNET and its variant. I think it is not enough.
> > >
> > > 3. For uncertainty, if you use dropout or model ensemble, I agree that they can be regarded as uncertainty. However, as mentioned, besides dropout and model ensemble, you also use "the average entropy of the per-voxel output distributions for single segmentation models", which can not be regarded as "uncertainty" from my view. I am also happy to continue to discuss with you the uncertainty estimation.

---

> > > > ### Author Response · Authors · 2022-11-18
> > > > **Response**
> > > >
> > > > 1. Thank you for acknowledging this contribution which you missed in your initial review. We also addressed your concerns about the medical data (it is not purely a compilation of public data as you claim) in our other comment. Hopefully, taken together, these points demonstrate that we have made substantial contributions with both datasets.
> > > >
> > > > 2. To be clear, UNET and UNETR are principally different architectures. A UNET is a symmetric model with convolution encoder and decoder, while a UNETR is a hybrid model with a Transformer encoder and a convolutional decoder. The UNET has a significantly higher level of built-in inductive bias than the UNETR. Both models are currently the de-facto standard in the medical field. If you feel that we should include a different architecture, can you please be specific about which one, and we will add it to the benchmark.
> > > >
> > > > 3. In established literature on uncertainty estimation (provided in in our previous response), there are two sources of uncertainty - data (aleatoric) uncertainty and knowledge (epistemic) uncertainty. Data (aleatoric) uncertainty is due to inherent noise, ambiguity and class overlap in the data itself. Knowledge (epistemic) uncertainty is due to a lack of understanding of the data by the model, which occurs where there is a distributional shift.
> > > >
> > > > If a model is trained with maximum likelihood, then an estimate of data uncertainty is captured by the entropy of a model's output distribution (such as the average per-voxel entropy). The _true_ level of data uncertainty is capture the the entropy of the true conditional distribution of the data. Knowledge (epistemic) is captured via measures of diversity of an ensemble, such as mutual information. Thus, in our work, the average per-voxel entropy of a single model is an estimate of data uncertainty, and can therefore be used to detect errors. This is entirely consistent with established literature.
> > > >
> > > > For additional details, please consult the above references as well as this one:
> > > > "Uncertainty estimation in deep learning with application to spoken language assessment", A Malinin, Phd Thesis (2019).

---

> > > > > ### Comment · Reviewer_1V51 · 2022-11-19
> > > > > **Thanks for your response**
> > > > >
> > > > > Thanks for your response.
> > > > >
> > > > > Through the rebuttal, the authors have added more information about the newly proposed datasets, including more explanation of the first medical image datasets and reminding of the new things of the second datasets. There are some different understandings of the model uncertainty, but there should not be a major concern, as uncertainty estimation is a challenging problem and is still widely studied.
> > > > >
> > > > > After acknowledging the new information about the proposed datasets, I will tend to maintain my original score "5: marginally below the acceptance threshold". The major concern is the limited or not enough contribution of introducing these two datasets. Maybe it is more proper to submit it to some other venues like "NeurIPS Datasets and Benchmarks Track".

---

### Author Response · Authors · 2022-11-07
**General Response 1**

Dear Reviewers, Area Chair,

We would like to thank reviewer 1V51 for raising an important point and address our response to all parties.

Reviewer 1V51 is right - this is first and foremost a dataset paper. This project came about after hearing from researchers, engineers and leaders across real-world industries who want to use AI to cure diseases, save the environment and make processes more efficient, but find that the models in the literature are designed for datasets which are simply not representative of the real world problems they are trying to solve. Nowhere is this more true than with the problem of distributional shift, which is omnipresent in real-world applications but very limited in standard datasets.

Distributional shift manifests in very different ways across different applications. The Shifts collection of datasets tries to capture a selection of tasks, each of which represents a category of problems that could be encountered in the real world, including:
- Predicting time-varying natural systems, with different initial conditions and seasonal patterns (Weather prediction task of Shifts 1.0)
- Predicting the trajectories of multiple interacting objects in a dynamic environment (represented by the self-driving car task in Shifts 1.0)
- Working with discrete, sequential natural language data. (Translation track in shifts 1.0)
- Interpreting 3D medical images in a very low-data regime (represented by the brain segmentation task in *this paper*)
- Predicting the performance of machinery given manual inputs in response to external conditions, subject to biased historical choices and changes to the performance profile over time, gradually and in response to specific events (represented by vessel hulls, subjected to speed and route choice, fouling and cleaning of the vessel power estimation task in *this paper*.)

In addition to allowing distributional shift to be examined in novel conditions relative to Shifts 1.0, making the last two datasets available solves an important problem for the community, as both domains (medical imaging and industrial optimisation) are notorious for difficulty in obtaining reliable datasets due to:
1. Difficulty and expense of data collection and
2. Difficulty of access due to complicated rights issues (sensitive medical data and commercially sensitive data respectively)

Obtaining this data required over a year of negotiation with various parties as well as extensive curation, compilation, analysis, preparation and expense on the part of our industrial collaborators and the original dataset owners. Succeeding in bringing all publicly available (to our knowledge) MS segmentation datasets under a single roof and releasing a vessel power prediction dataset of this kind for the first time are major milestones.

Thus, while there are multiple medical datasets and tabular datasets available, the value in our work is that we provide easy access to hard-to-get industrial and medical data with unique properties that has been partitioned appropriately (as well as consistently pre-processed) to investigate how well models perform under distributional shift - something which is of key importance in industrial applications.

We are confident the datasets presented in this paper can bring significant value to the ML community. The data, code and the baselines are already publicly available and the leaderboards are running - this value is already out there for people to access. A publication at ICLR will enable us to draw the attention of the ML community to this data, increasing its potential impact and allowing more researchers to know about, access and benefit from this valuable resource.

Kind Regards,
The Authors

---

> ### Comment · Reviewer_1V51 · 2022-11-18
> **Thanks for your response**
>
> Thanks to the authors for the rebuttal.  As a machine learning researcher in the medical image field, I think I can understand the importance and impact of adopting ML/AI to solve healthcare problems, and I can fully understand the motivation for building this benchmark dataset. I have no doubt about the motivation of the authors, and I fully appreciate the author's effort to work in this field.
>
> However, the rebuttal and contribution of this paper seem "overclaimed". Following are my own thoughts.
>
> 1.  I totally agree that building a benchmark to evaluate different algorithms for tackling domain shifts is very important. At the same time, I need to mention that the contribution of an individual paper should be **ONLY** evaluated from the paper itself. As mentioned by the authors,  They have published a paper Shifts 1.0 and introduced four different datasets. In this NEW paper, they introduce two additional datasets to enhance the original datasets. My concern about the contribution is about the **marginal contribution** of this paper. I have no doubt about the contribution of the first Shifts 1.0 paper. Actually, I think the original Shifts 1.0 is a very good paper and worth the credit. But I have doubts about whether the contribution of introducing **two additional datasets** can meet the contribution of the ICLR paper, especially based on the published paper Shifts 1.0. Maybe the AC or other reviewers can make judgments.
>
> 2. The authors argue the difficulty of acquiring medical data. "Obtaining this data required over a year of negotiation with .... " I totally agree with that, and this is also the reason why releasing high-quality medical image datasets can have a bigger contribution and can be published in a high-quality journal. Unfortunately, as I have previously mentioned, the authors DID NOT release their own datasets but collected the public datasets together. In this sense, I am not sure about the contribution and difficulty of that.
>
> 3. The authors argue that "Thus, while there are multiple medical datasets and tabular datasets available, the value in our work is that we provide easy access to hard-to-get industrial and medical data with unique properties that has been partitioned appropriately (as well as consistently pre-processed) to investigate how well models perform under distributional shift - something which is of key importance in industrial applications." However, I am not sure if this work is **really** easy-access. The previously mentioned medical image datasets are both publically available and actually used for medical image analysis challenges.  Also, the authors mentioned, "The data, code and the baselines are already publicly available and the leaderboards are running - this value is already out there for people to access." However, I have not found this public data, code, or public leaderboards. Maybe I missed them. Please point out if I miss them.

---

> > ### Author Response · Authors · 2022-11-18
> > **Response to Rev 1V51**
> >
> > Dear Reviewer,
> >
> > ### Disclaimer
> >
> > We have made no claim to be the authors of Shifts 1, neither in the paper, nor in any previous responses. We are only claiming to be working within the greater context of Shifts (the public benchmark). In our prior response we aimed exclusively to contextualise what Shifts  aims to do, and how we expand upon that. Your claim that we are the authors of Shifts 1 violates the rules of this double-blind rebuttal process. Similarly, we cannot provide you with any external links to the leaderboard, code, models and other resources, as that would also de-anonimize us and violate the rules of the double-blind review process. Additionally, we cannot encourage you to search for these resources. We have only mentioned them because the two other reviewers asked about the status of such resources. Whether we are, or are not the authors of Shifts 1 should not be a point of discussion in a double-blind review.
> >
> > That being said, we do appreciate that the nature of the work (dataset paper) makes it difficult to strictly maintain the double-blind nature of this rebuttal.
> >
> > ### Addressing your concerns
> >
> > 1. We agree that any paper should stand on its own. Whether a paper is sufficiently good for publication at ICLR is an object of subjective discussion. In this context, we can only point to past precedent. For example, last year at ICLR 2022 the paper "Extending the WILDS Benchmark for Unsupervised Adaptation", which was an extension to the original WILDS dataset paper, was considered so good by the AC that it was accepted as oral with a comment that "Vision, Clarity and Significance are clearly above the bar of ICLR". This paper added  unsupervised data to a subset of the WILDS datasets to enable the study of unsupervised domain adaptation.
> >
> > 2. It is inaccurate to state that our work is simply compilation of existing published datasets re-released under a new name. We have contributed the following:
> >     - Provided access to a **new and unpublished** MS dataset, collected at Lausanne. Researchers can evaluate their models on this data via the public leaderboard we have constructed.
> >     - Released **new and unpublished** marine vessel power consumption data, including the introduction of the synthetic generalisation set, which enables researchers to study causal disentanglement, which you acknowledge is a contribution.
> >     - In addition to providing easier to access to previously published medical data (we address this point below), we have also provided an appropriate partitioning that enables the study of robustness to distributional shift in a risk-critical application of machine learning. The creation of this partitioning was made possible only by combining the different existing datasets, which are different (shifted) from each other in various attributes. Designing this partitioning and verifying its appropriateness to study distributional shift was non-trivial.
> >
> > 3. Of the previously published datasets which we used, both ISBI and PubMRI are open-access and freely available. However, obtaining the MSSEG data can take **up to several months**, because each project needs to be independently reviewed by OFSEP a monthly meeting. This process need to be undertaken by _each_ researcher or challenge participant, which in practice limits how many people work on this data or participate in public challenges. Thanks to our work and negotiated agreement with OFSEP, all researchers can **now obtain this data very quickly within a day**. **In the last 60 days around 120 researchers have accessed our data**, which is readily available online. We cannot provide the link here in this rebuttal, due to anonymity constraints.

---

### Author Response · Authors · 2022-11-18
**Manuscript Updated**

Hello!

We have updated the manuscript with the following changes:

- We have added references to different uncertainty estimation techniques and commented that they do not impact the model's predictive performance. This is added to the last paragraph of section 2 in the main paper.
- We have also added very short discussion of how we have selected the splits and that location is a proxy for multiple shifts, not just scanner type. This is added to the paragraph below table 1 in the main paper.
- Sub-population analysis done and updated the paper. The sub-population analysis is provided in appendix C.5 of the supplementary (file also updated).

Kind Regards,
Authors

---

### Author Response · Authors · 2022-11-19
**Summary**

A brief summary of the paper, reviews and our rebuttals and updates to the manuscript.

Our paper introduced two novel tasks and associated data for evaluating deep learning models under conditions of in-the-wild distributional shift. They aim to contribute to the community's efforts to apply deep learning to more diverse applications, more fairly evaluate the utility and safety of models for downstream tasks and gain greater understanding of the properties and robustness of neural networks. A novel vessel power prediction task is released, alongside a first-of-its-kind dataset from real vessels, combined with a synthetic dataset generated to enable the causal disentanglement properties of models to be evaluated. A series of public, but previously difficult to obtain MRI brain scan datasets are combined into a new, instantly available public dataset and partioned to enable distributional shift properties of models to be evaluated, with a wholly new, previously unavailable brain scan dataset made available for model evaluation.

Reviewer cD8F recommended accepting the paper and asked us three clarification questions, which we answered to their satisfaction and they confirmed they are maintaining their acceptance recommendation.

Reviewer AFoe recommended accepting the paper, but requested additional discussion and analysis of fairness and the sources of shifts in the MRI data. We conducted a sub-population analysis and added it to the paper, along with a discussion of the sources of shifts and split selection.

Reviewer 1V51 recommended rejection citing five basic concerns, each of which we addressed as follows:
1. That the medical image data were already public and thus not novel => We pointed out that i. we made available a new dataset for model evaluation which had not previously been public ii. the public data we combined into the other datasets had previously been very difficult to obtain (requiring individual permissions taking several months) while we made it instantly accessible, and iii. we introduced a novel partitioning for the public data based on analysis of the metadata which enables it to be used to evaluate distributional shifts, which is the core object of the work.
2. That the vessel power dataset was too similar to the weather prediction dataset and thus not novel => We explained the novelty of the vessel power prediction task and, in particular, the significance of the generation of a synthetic data set for evaluating causal disentanglement alongside the real data, which Reviewer 1V51 accepted they had underestimated in their initial assessment and in fact constitutes a contribution.
3. That we only evaluated one benchmark model on the medical data => We pointed out that we evaluated two principally different architectures (UNET and UNETR), which constitute the de facto standard in the field. We asked the reviewer to suggest other models we should add to the benchmark but a response was not forthcoming.
4. That we had not cited sufficient works around uncertainty => We added these citations to the paper
5. That the reviewer was not able to understand what we meant by uncertainty => We provided a detailed explanation of the concepts of aleatoric and epistemic uncertainty and how they relate to the paper in our rebuttal and added the relevant discussion to the paper

---

> ### Comment · Reviewer_1V51 · 2022-11-19
> **Some Comments**
>
> It is important to maintain a good environment for discussion. Using some language or expression "tricks" to "overclaim" something may not be encouraged.
>
> 1. My original concern is about the limited contribution of this paper, not claiming no contribution to this paper. Building a benchmark is important and should be encouraged, which is also acknowledged by my previous comments. However, my main concern is the **limited contribution or maybe not enough contribution** extending the Shifts by including two additional datasets. The authors argue that the ICLR 2022 paper "Extending the WILDS Benchmark for Unsupervised Adaptation" is also an extension of existing datasets. I have briefly checked this paper. It seems that the authors ignore the solid effort of this paper by extending 8 different datasets. "We present the WILDS 2.0 update, which extends 8 of the 10 datasets in the WILDS benchmark". Therefore, I still think the contribution of this paper cannot meet the standard of ICLR, even taking the author's newly argued contribution as input.
>
> 2. For the newly introduced medical image dataset in Lausanne. I need to mention that it is only provided for evaluation via docker. That means the users cannot access the raw image and annotations for the new dataset and can only submit a model for evaluation.  In this case, it is hard to argue that this dataset is **published**.
>
> 3. The authors argue that "other datasets had previously been very difficult to obtain (requiring individual permissions taking several months) while we made it instantly accessible" However, no statistics or data can support this claim, and this is only a subject description.
>
> 4. The authors also argue that "we have also provided an appropriate partitioning of original datasets". This new partitioning is "When multiple scans are available per patient, we ensure that all scans for a particular patient appear only in one dataset." However, this is common sense and widely recognized when conducting medical image analysis to avoid data leakage. When dividing the training, validation, and testing sets for medical image analysis, we will follow patient-level partitioning, not image-level partitioning.  In this case, I cannot agree that this partitioning strategy can be regarded as new and a contribution.
>
> 5. UNET and UNETR are strong baselines for medical image segmentation, but there are also some other widely adopted architectures for medical image segmentation, like the advanced nnUNET, UNet++,  TransUNet (the first work of incorporating transformer backbone into U-Net architecture). For the mentioned MS segmentation tasks with many small lesions, dilated convolution-based algorithms should also be considered.
>
> 6. I am not an expert on uncertainty estimation, but I have several related projects where we employ model uncertainty as a tool. I think I have some basic knowledge of model uncertainty. But of course, I may have some misunderstanding about that. My original concern is that uncertainty should somehow involve "the difference or comparison with multiple prediction results", like using different data augmentation to calculate different prediction results or using model dropout to calculate different results.  I also understand that entropy can be regarded as an uncertainty metric, but it seems that it should be calculated using the averaged prediction probabilities of multiple outputs, which is different from the average per-voxel entropy of a single output. If there are some references that support per-voxel entropy as uncertainty, it is good to mention them directly (for example, the specific paper or paragraph). Listing a bunch of papers to let the reviewers "learn" them may not be a good practice.

---

> > ### Author Response · Authors · 2022-11-19
> > **Further Response Pt 1**
> >
> > 1. We don't want to insist too much on comparison to other papers, but it is important to mention for completeness that all 10 datasets in WILDS 1.0, and all 8 additions that WILDS 2.0 makes to WILDS 1.0 are adapted from existing, published public datasets shared under very permissive licenses. The critical contribution of WILDS (1 & 2) is setting up these datasets, pre-processing and repartitioning them to enable the evaluation of robustness to various forms of distributional shift. We have published one entirely new corpus of supervised data from a domain where obtaining data is very difficult, made available for evaluation a new dataset of brain scans and compiled together, repartitioned and released in a more accessible format further brain scan data sources. Obtaining permission from the various original collectors of the medical and shipping data took significant diplomatic effort and time. We leave it to the AC to determine whether this is sufficient contribution.
> >
> > 2. We agree that this is limited, but such the is the difficulty with sharing medical data. The data was originally collected with a clause stating that study participants can have their (individual) data removed at any time after the study. There is no way that the original data collectors (who we work with) can share the raw data publicly and still hold to the provisions under which the data was collected. Providing dockerized access is the only way to make this data publicly available. This is a typical example of the difficulties of obtaining, sharing and working with medical data.
> >
> > 3. It took us over two months to originally obtain the MSSEG data. Obviously, this is anecdotal evidence, but everyone we spoke two who tried to obtain the MSSEG data from the Shanoir platform had a similar experience.
> >
> > 4. This is **not accurate**. The medical partitioning is described in table 1. This is a **five-fold** partitioning, with a training set, a dev and eval set which are **matched** to the training data, and a dev and eval set which are **shifted** relative to the training data. Furthermore, the shifted dev and eval are also shifted **relative to each other**.  The distributional shifts are by **location where the data was collected**. The training and matched dev and eval data contain data from Rennes, Lyon, Bordeaux and Best. The Shifted Dev data comes from Ljubljana, and the shifted eval data comes from Lausanne. Location serves as a proxy for multiple latent shifts, including scanner type and acquisition protocol, disease prevalence, ethnicity, age and stage of MS. Please examine our response to reviewer AFoe. Furthermore, we conducted a study of between which locations the distributional shift is the greater. This examination is detailed in section C.1 of the supplementary. This 5-fold split is what allows investigation of robustness to distributional shift. A 5-fold partitioning is also used for the marine cargo vessel power consumption data.
> >
> > 5. Thank you for the suggestion. We will include these baselines in the next version of the paper.

---

> > > ### Author Response · Authors · 2022-11-19
> > > **Further Response PT2**
> > >
> > > 6. What you refer to is knowledge uncertainty, also known as epistemic or model uncertainty. This one of the two sources of uncertainty. You are correct in that knowledge uncertainty is estimated by examining the diversity of multiple predictions from the same input. This is done via ensembles (of various forms), test-time data augmentation or having multiple independent output layers.
> > >
> > > However, data uncertainty, also known is aleatoric (dice-throwing) uncertainty is not a property of the model, but of the underlying distribution of data itself. Consider that the data is produced by the following distribution (that we do not have access to):
> > >
> > > $$
> > > (y^{i}, x^{i})_{i}^N =  p_\{tr\}(y,x)
> > > $$
> > >
> > >
> > > Here, the **true** level of data uncertainty for a given input X is the entropy of the true marginal:
> > > $$
> > >  H[ {\tt p}_{tr} (y | x) ]
> > > $$
> > >
> > > And the overall level of data uncertainty is then by the average entropy of the conditional over all x. This is the true underlying level of data uncertainty. It represents class overlap, similarity of different categories (dogs and wolves, for example, as opposed to dogs vs cats), or noise, such as sensor or measurement noise. In the context of medical image segmentation, data uncertainty is high at lesion boundaries. For the ship data, this is sensor noise.
> > >
> > > Given a probabilistic model $ {\tt p}(y | x, \theta)$ which is trained on this data via maximum likelihood, which is equivalent to minimising the KL divergence between the model $ {\tt p} (y | x, \theta)$ and the true underlying conditional distribution of data ${\tt p}_{tr} (y | x)$ , the conditional entropy of the model $H[  {\tt p} (y | x, \theta) ]$ will be an **estimate of data uncertainty**. The quality of this estimate is conditional on the power of the model, whether it is converged or not, whether it has overfit or not and, for regression tasks, whether the chosen output distribution (Normal, Student-T, Exponential, etc...) is the same as that of the distribution which generated the data.
> > >
> > > For the medical task, the average per-voxel entropy of a single model is an estimate of data uncertainty, and has value is detecting errors in regions where data uncertainty is high, such as lesion boundaries.

---

> > > > ### Comment · Reviewer_1V51 · 2022-11-19
> > > > **Thanks for your response**
> > > >
> > > > 1. For the uncertainty estimation part, thanks for your detailed explanation.  As previously indicated, I am not an expert on that, so I left the assessment to AC or others.
> > > >
> > > > 2. For other points, thanks for your further clarification. I think it is more clear now to understand the scope of this paper. Assessing the "contribution" or "novelty" is a subjective thing. As a reviewer, I express my own assessment and will maintain my original score considering the not enough contribution from my view.

---

### Decision · Program_Chairs · 2023-01-20

**Decision:**

Reject

**Justification For Why Not Higher Score:**

Limited number of datasets; the medical dataset based on public data; limited contributions relative to the original Shifts paper.

**Justification For Why Not Lower Score:**

N/A

**Metareview: Summary, Strengths And Weaknesses:**

The authors propose two new datasets for measuring distribution shift. This works builds upon the earlier work of "Shifts 1.0". One of the datasets is a medical dataset, which the authors took a painstaking amount of time to gather (although the data itself was already public); while the other is vessel power estimation. The reviewers noted that creating benchmarks is an important problem; however, they noted the more modest contribution compared to "Shifts 1.0".

The deciding factor, however, was that the reviewers were unable to look at even at a sample of the dataset. While such datasets could be a boon to the community, but unfortunately, without reviewer scrutiny it's hard to know if the dataset contained errors.

**Summary Of Ac-Reviewer Meeting:**

During discussion, it became clear that everyone was lukewarm about the paper. The one person who voted for accept (with a score 6) mainly did so because of the difficulty of creating medical datasets. What became clear during the discussion was that even though they asked the authors for sample data, they were not given it due to anonymity issues. We all agreed in the end that the paper would be improved with a revision and submitted to a future conference.